# Consumer attitudes and perceptions on consumption of edible insects among communities in western Kenya

Emmah Owidi[1], Gilbert Asoka[2], Eric Waga[3], Alfred Ochieng'[4], Fanuel Kawaka[4]*

**1** Department of Environmental Education, Kenyatta University, Nairobi, Kenya, **2** IPSOS Kenya, Nairobi, Kenya, **3** Independent Research Consultant, Nairobi, Kenya, **4** Department of Biological Sciences, Jaramogi Oginga Odinga University of Science and Technology (JOOUST), Bondo, Kenya

* fkjairo@hotmail.com

## Abstract

Edible insects are a highly sustainable and nutritional food source despite their low consumption in many communities. This study evaluated consumer attitudes and perceptions on consumption of edible insects in western Kenya. Eight focus group discussions (FGDs) were conducted in four rural and four urban markets in Kisumu and Vihiga counties. The FGDs were audio-recorded, transcribed, and analyzed using thematic content analysis. A total of 59 respondents consisting of 27 male and 32 females participated in the FGDs. The results showed higher acceptability and consumption of edible insects in rural areas compared to urban centers. The most popular edible insects were flying termites, grasshoppers, soldier termites and locusts. Consumption of these insects was mostly determined by their perceived high nutritional value, pleasant taste and familiarity as food sources. Seasonal capturing of the insects, especially during off-season also contributed to their low consumption. Respondents' willingness to taste insect-based products was based on curiosity, packaging and familiarity with the product forms. These results suggest that there is need for interventions that promote public awareness and enhance the availability of edible insects and insect-based products, to support wider acceptability and consumption.

## 1. Introduction

The increasing global human population, food waste, climate change, unsustainable human practices, and changing consumer food preferences worsen food and nutrition insecurity in developing countries [1]. The increasing cost of animal protein production and rising environmental pressure on agriculture and livestock farming have necessitated the search for sustainable alternatives [2]. Innovative techniques for food production should take into account the nutritional, environmental, and socio-cultural dimensions of food sustainability [3]. Consequently, international humanitarian organizations such as the Food and Agriculture Organization and the United Nations have urged for the promotion of alternative and sustainable food sources including insects [4,5]. Consumption of insects as human food has the potential to meet the above demands and prove to be a valid strategy for improving

**Data availability statement:** All relevant data are within the manuscript and its Supporting Information files.

**Funding:** This study was supported by a grant from the Nestlé Foundation for the study of the problems of nutrition in the world, Lausanne, Switzerland.

**Competing interests:** The authors have declared that no competing interests exist.

global food security [6,7]. Insect production has a higher conversion rate to food compared to conventional livestock and can be grown on organic waste, occupy less production space, and produce less greenhouse gases [8]. Edible insects are a healthy source of food with high quality fat, protein, vitamin, fiber, and micronutrient content for humans [9,10]. Insect consumption is currently receiving substantial attention as a potential food source of high nutritional value, with important environmental benefits [11]. There are many benefits associated with the consumption of insects as an alternative to conventional food animals [12,13]. Insects are extremely important in terms of ensuring global food, nutrition, and feed security [14]. In addition, edible insects are considered safe for human consumption and extremely beneficial for high-quality diets [15].

In Kenya, different insect species have been traditionally consumed by many communities, especially in the western regions of the country [16,17]. However, many consumers still consider insects an unappealing and disgusting food source, despite edible insects forming a part of traditional diets in local households. The apparent viability and growing interest in edible insects as a sustainable and alternative source of food still face many barriers that impede their widespread uptake [11]. Several studies have emphasized the need to examine the underlying consumer attitudes toward the consumption of edible insects [18,19]. However, few attempts have been made to evaluate the consumer attitudes, perceptions, and acceptability of the consumption of edible insects. This study investigated the attitudes and perceptions of local communities towards a range of edible insects and insect-based products in western Kenya.

## 2. Methods

### 2.1. Study site

The study was conducted in 8 markets/trading centers (4 rural and 4 urban) in Kisumu and Vihiga counties of western Kenya. The rural markets included Emabungo, Ekwanda, Depo, and Lela, while urban markets included Daraja Mbili, Chulaimbo, Nyawita, and Maseno. The rural markets represented areas where edible insects are easily available, accessible, and form part of local household diets. The urban markets represented areas where insects are sold and consumed but are not considered a main source of food. The study site was selected due to the prevalence, sale, and consumption of edible insects among local communities [20–22]. Vihiga County is composed of communities that predominantly speak the Luhya dialect while Kisumu County consists of predominantly Luo-speaking communities. However, some markets such as Nyawita were situated within the boundaries of the two counties and hence were constituted of respondents from both communities.

### 2.2. Study design and respondents

The study used focus group discussions (FGDs) to collect qualitative data from the consumers. The choice of the method was due to its wide use in qualitative investigation in the field of nutrition and food technology [23,24]. Similar studies have adopted the same data collection technique in exploring consumer attitudes, opinions, and perceptions towards certain food products [24,25]. Adult individuals of mixed genders, age, education levels, and occupations were recruited for study participation through referral by community gatekeepers including local administration officers (chiefs and village elders).

### 2.3. Data collection

The FGDs were conducted between 1st of October and 30th of November 2017 in each of the 8 markets, with each FGD consisting of 7–9 respondents. FGDs were constituted to represent two

groups of individual consumer experiences: insect eaters and non-insect eaters. The FGDs were conducted in private spaces in each market by two experienced qualitative researchers (a male and a female) in Kiswahili, Luhya, and Luo languages, depending on the respondents' preferences. A semi-structured topic guide was used, and the discussions were audio-recorded with the respondents' permission. At the beginning of each FGD, respondents set ground rules with the guidance of the moderators, to minimize interruptions and encourage respectful interactions. Respondents were anonymized using pseudonyms, e.g., 'R1' (Respondent 1) to assure them of confidentiality, and moderators maintained a non-judgmental attitude when asking questions to promote open and honest feedback. Quieter respondents were encouraged to share their opinions and more dominant respondents encouraged to allow alternative views. Each FGD lasted approximately 2–3 hours and sessions were divided into 4 stages of discussion topics as described by Tan et al. [25]. These included individual experiences and knowledge of edible insects, reasons for consuming or not consuming edible insects, knowledge and perceptions of specific edible insect species using images, and knowledge and perceptions of insect-based products. Respondents were also requested to vote for their most and least liked edible insect species using displayed images, and results were shared on flipcharts for discussion.

## 2.4.  Data analysis

Audio recordings from focus group sessions were transcribed and translated into English. Notes recorded by note-takers including nonverbal expressions and actions during FGD sessions were used to verify the data. Individual responses were recorded and checked for accuracy. The evaluations on insect species and insect-based products were coded according to positive or negative consumer perceptions, and the codes were grouped by the themes being evaluated. The ratings on the expected liking for each of the selected insect species were averaged within respondent groups. Thematic content analysis was used to analyze qualitative data inductively and deductively [26]. The NVivo software 12 (QSR International Pty Ltd, 2021) was used for coding de-identified FGD transcripts. Author EW reviewed all the transcripts, generated key themes, and developed a codebook based on the FGD guide topics. Authors EW and EO then selected a subset (n = 2) of the transcripts and double-coded them to ensure coding consistency. The authors reviewed the codes continuously and resolved discrepancies over regular meetings until an agreement was reached. Author EW independently coded the remaining transcripts and grouped the emerging themes from the interviews. Once coding was complete, author EO summarized and organized data by the key themes of the discussions and by respondent types, either as rural or urban; edible insect eaters or non-eaters, and compared the findings between the different groups [25,27].

## 2.5.  Ethical statement

Ethical approval was granted by the Ethics and Research Committee (ERC/EXT/001/2016) of Pwani University. All the study respondents provided informed consent before study participation, and all the study procedures were conducted as per the guidelines and regulations of the National Commission for Science and Technology and Innovation (NACOSTI), Kenya. All participants were above 18 years old and provided written informed consent before study participation and were given a copy of the informed consent documents for their records.

## 3.  Results

### 3.1.  Community experiences and knowledge of edible insects

A total of 59 respondents consisting of 27 male and 32 females participated in the FGDs. All the respondents in the eight markets identified the commonly consumed edible insects in

their communities. About seven types of common edible insects were identified (Table 1). These insects including flying termites, grasshoppers, locusts, queen termites, soldier termites, edible black ants, and crickets were typically consumed raw, salted, roasted, fried in a pan without oil, or sun-dried, as part of a meal or snack. Most respondents identified the insects using the local dialects while a few were identified in English and Kiswahili, Kenya's official languages.

Flying termites were the most known and liked edible insects identified in all eight markets (Fig 1). Identification of different flying termite species was based on the times the insects came out, e.g., at night (*Agoro*) or afternoon (*Sisi, Ng'wen, Oyala*), the color, number of wings, size, and shape. Additionally, respondents reported knowledge of identifying the edible species, when, where, and how to collect them, and how to prepare them for consumption. The second most common edible insects were grasshoppers (*Odede, Senene, Ongogo*), and soldier termites (*Biye, Ching'eng'eni*), with each identified in five markets (two rural and three urban). Desert locusts (*Nzige, Bonyo, Chisiche*) were identified in three rural markets, crickets (*Onjiri, Esirali*) in two urban markets, and queen termites (*Nyailiel, Omwami*) in one urban market. Edible black ants (*Onyoso*) were also indicated to be popular and were identifiable by their large abdomen. The knowledge of edible insects was similar across the markets; however, the acceptability and consumption varied between rural and urban markets. Other knowledge

**Table 1. Common edible insects identified by the FGD respondents.**

| No. | Order | Common name | Local names | Local language |
|---|---|---|---|---|
| 1 | Isoptera | Flying termite | *Agoro* | Luo |
| | | | *Aming'aming'* | Luo |
| | | | *Ching'eng'eni* | Luhya |
| | | | *Chiswa* | Luhya |
| | | | *Kumbe kumbe* | Swahili |
| | | | *Ng'wen* | Luo |
| | | | *Oyala* | Luo |
| | | | *Ogao* | Luo |
| | | | *Sisi* | Luo |
| | | | *Amaabuli* | Luhya |
| | | | *Amafwetere* | Luhya |
| 2 | Hymenoptera | Black ant | *Onyoso* | Luo |
| 3 | Isoptera | Soldier termite | *Biye* | Luo |
| 4 | Orthoptera | Desert locust | *Bonyo* | Luo |
| | | | *Chisiche* | Luhya |
| | | | *Nzige* | Swahili |
| 5 | Orthoptera | Grasshopper | *Odede* | Luo |
| | | | *Senene* | Luhya |
| | | | *Ritete* | Luhya |
| | | | *Ongogo* | Luo |
| 6 | Isoptera | Queen termite | *Nyailiel* | Luo |
| | | | *Omwami* | Luhya |
| 7 | Orthoptera | Cricket | *Onjiri* | Luo |
| | | | *Esirali* | Luhya |
| | | | *Esichiriri* | Luhya |
| | | | *Lichiriri* | Luhya |
| | | | *Litakala* | Luhya |

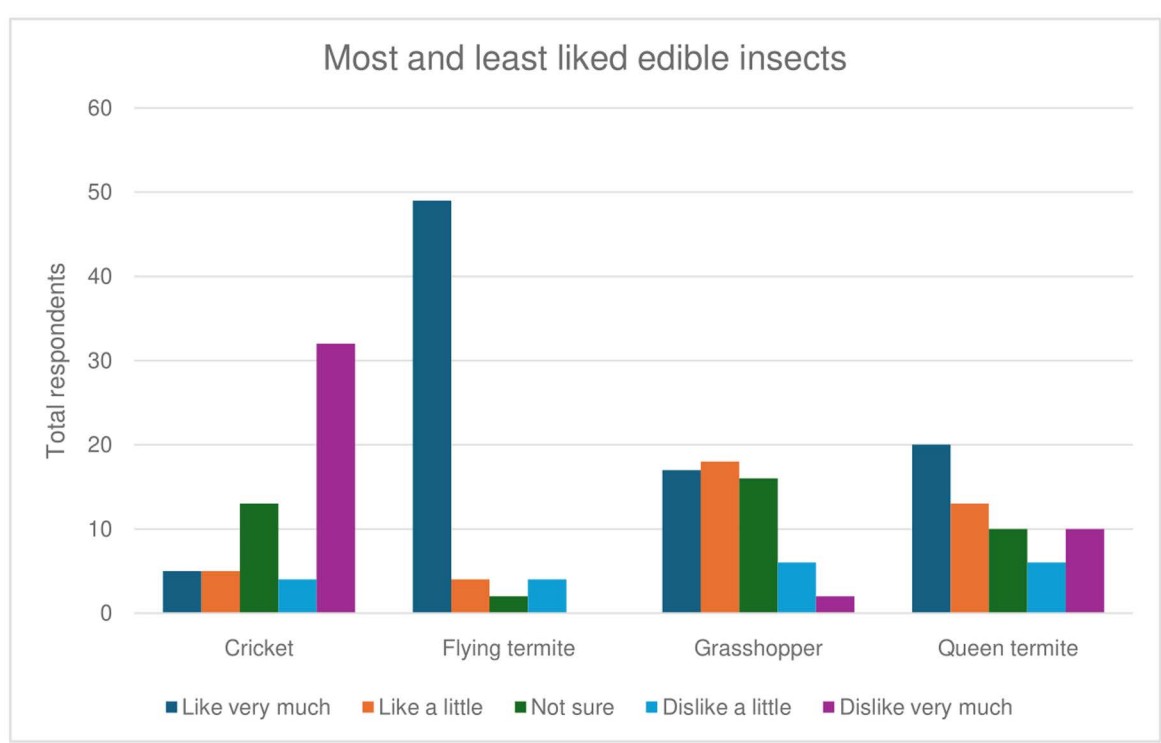

**Fig 1. The most and least liked edible insects among FGD respondents.**

was related to the seasonal occurrence of edible insects, especially flying termites which were reported to mostly occur during rainy seasons. However, respondents indicated that some communities had devised traditional methods of directly harvesting flying termites from anthills beyond rainy seasons.

### 3.2. Factors influencing consumption of edible insects

Nutritional value, taste, and familiarity of edible insects as a source of food were the most important determinants of edibility among respondents (Table 2). Moreover, respondents considered *"appearance"* and *"tastiness"* as key factors in deciding whether to eat or not to eat edible insects. Individuals who consumed edible insects (eaters) described them with words such as *"tasty"*, *"sweet"*, *"delicious"*, and *"appetizing"*; while non-eaters described them as *"disgusting"*, *"can make one vomit"*, *"contains pus"*, *"ugly"*, and *"don't look good"*. Other respondents cited the high nutritional value of edible insects especially proteins, fats, and minerals like iron as the reason for their decision to include them in their diet. Similarly, certain edible insects' medicinal and health benefits were cited in two rural and two urban markets as an incentive for their consumption. Respondents reported that most knowledge on the benefits of edible insects has been traditionally passed down within communities. For example, a respondent in one rural market narrated that her mother had informed her that eating flying termites (*Ng'wen*) mixed with soldier termites (*Biye*) prolonged life, while a respondent in an urban market indicated that eating queen termites (*Omwami*) contributed to increased breast milk production. Queen termites were also described as helpful in decongesting the chest in asthmatic patients while eating flying termites aided in detoxification by cleaning the digestive system, especially in children.

**Table 2. Reasons for the consumption of edible insects among FGD respondents.**

| Reason for consumption | Markets | | | | | | | |
| --- | --- | --- | --- | --- | --- | --- | --- | --- |
| | Urban | | | | Rural | | | |
| | Daraja Mbili | Maseno | Chulaimbo | Nyawita | Emabungo | Depo | Ekwanda | Lela |
| 1. Good taste and boosts appetite | X | X | X | X | X | X | | X |
| 2. Nutritional value | X | X | X | X | X | X | X | X |
| 3. Source of food and satisfies hunger | X | X | X | | X | X | X | X |
| 4. Medicinal and health benefits | | X | | X | | | X | X |
| 5. Naturalness | | | | | | X | X | X |
| 6. Influence from others | | X | | X | X | | X | |
| 7. Low cost or affordable | | X | X | | | | X | |
| 8. Easily available | | | X | | | X | | |
| 9. Cultural significance | | | X | X | | | | |
| 10. Curiosity | | | | | | X | X | |

*"Most of them [edible insects] are sweet and nutritious…When you fry Kumbe kumbe [flying termites], you can eat it with ugali, or you can even take it with tea. You can even eat them raw or when they are cooked." (FGD Maseno, urban)*

*"The reason why we like to eat these insects [is], a long time ago my mother told us that if you eat Ng'wen [flying termites] and you mix with termites…biye you live for long, you cannot die quickly. That is why we love [them] so much." (FGD Lela, rural)*

The lack of chemical additives was also cited as one of the reasons for consumption of edible insects. In seven markets, respondents indicated that they ate insects simply because they considered them as food or to satisfy hunger while others indicated that edible insects served as an accompaniment to other meals or as snacks. Respondents in four markets indicated that their decision to consume edible insects was influenced by other people, especially family members such as grandparents, parents, and in-laws. Affordability and curiosity were also cited to influence insect consumption, with edible insects being easily available and not costly during certain seasons, hence were accessible to people of low-income levels who could not afford other protein sources. When collected in surplus, insects such as flying termites were sold in local markets to generate income and were liked by people across different social and income groups. Furthermore, certain insects also had cultural significance, for instance a type of green grasshopper locally known as *'Senene'* was considered a very special meal that was typically prepared for very important guests such as sons-in-law in some communities. Similarly, queen termites were considered prestigious and were reserved for the male household head or the oldest person in the home as a sign of respect.

### 3.3. Reasons for non-consumption of edible insects

Respondents noted that not all community members consumed edible insects due to reasons such as unfamiliarity and lack of prior knowledge or consumption experience. Particularly, people who were not native to the study areas or who grew up in areas where edible insects were not typically available or consumed found it harder to accept them as a food source. For example, women who were married into the area from other communities attributed their inability to eat edible insects to having grown up in areas where they were not consumed. It was also noted that younger people, especially in urban markets, did not have much knowledge about edible insect species hence did not consider them as food sources. In such cases,

the loss of traditional knowledge on how to find and collect edible insects was also noted. Respondents reported that although edible insects such as ground crickets were usually collected by the older men who considered them a special delicacy, the insects were rare and their collection was difficult, discouraging their use as food. Similarly, collection of queen termites was done by adult men and required specific expertise and physical strength. Flying termites were mostly collected by women and children, but their collection was considered difficult. Collection and consumption of grasshoppers was done by children who found collecting them to be fun, and elderly people who considered them a vital component of their diet as a substitute for meat and for perceived nutritional benefits such as strengthening of joints. However, adults found catching grasshoppers to be challenging and ridiculous.

> *"…Getting it [queen termite] is like getting gold because I think it is found in an anthill. You must dig and you must time it. You'll get soldiers in there, army commanders, which will bite you before you reach this insect…" (FGD Depo, rural)*

*"Disgust"* and *"ugliness"* were cited as the most important factors influencing decisions to not consume edible insects (Table 3). Most edible insects were considered disgusting and could make people vomit. The physical appearance of edible insects also informed their choice as food. For instance, insects perceived as being *"too fatty"* or *"containing pus"* were less favored or considered unpalatable by non-eaters. Particularly, black ants (*Onyoso*) were described as ugly because of their large abdomen, while crickets and queen termites were considered *"scary"*. Although insect eaters simply considered edible insects as food, most non-eaters did not, instead perceiving them as pests, or otherwise non-beneficial or harmful insects. Cultural and religious beliefs were also cited as contributors to the non-consumption of edible insects. Some communities believed that edible insects came from places where dead people were buried while others believed they caused harmful effects such as deafness, which was mentioned for *'Aming'aming'*, a flying termite species. Several respondents indicated that some religious sects, such as *Legio Maria*, forbid the consumption of edible insects. Other reasons for non-consumption of edible insects included fear of physical allergic reactions such as

**Table 3. Reasons for non-consumption of edible insects among FGD respondents.**

| Reason for non-consumption | Market Location | | | | | | | |
|---|---|---|---|---|---|---|---|---|
| | Urban | | | | Rural | | | |
| | Daraja Mbili | Maseno | Chulaimbo | Nyawita | Emabungo | Depo | Ekwanda | Lela |
| 1. No prior knowledge or experience | X | | | X | | | | |
| 2. Perception as pests or non-beneficial insects | X | | | | X | | | |
| 3. Religious restrictions | X | | | | X | | | |
| 4. Dislike and disgust including vomiting | X | | | X | X | X | X | |
| 5. Cultural beliefs and taboos | X | X | X | X | X | X | X | X |
| 6. Unattractiveness | | | | | X | X | X | |
| 7. Physical composition | | X | X | | X | | X | |
| 8. Allergy and side effects | | X | X | | X | X | X | |
| 9. Method of preparation | | | | | | X | X | |
| 10. Socio-economic status | | | | | | | | X |
| 11. Health and hygiene considerations | | | | X | | X | | |
| 12. Fear of bites or injury | | | | | | | X | |
| 13. Small sizes of insects | | | | X | | | | |
| 14. Influence from other people | | X | | | | | | |

stomach upset, diarrhea, and rashes. Methods of preparation also influenced consumption, with some respondents indicating they could only eat edible insects in fried rather than raw form. Socio-economic status was also reported to influence non-consumption, as collection and consumption of edible insects was associated with poverty, being uneducated, or idleness, and was considered primitive, hence was embarrassing for some people in the community. Therefore, households of a high social status either bought edible insects already collected by poorer people from the local markets rather than collecting them themselves, or depended on alternative diets and shunned consuming insects, leaving them for poorer households.

*"I eat them because other people are eating them, but I don't like them that much because they don't look good. When you press them, they remove pus and that's why I don't like eating them." (FGD Ekwanda, rural)*

*"Some believe that Kumbe kumbe are those who died long ago that is why they refuse to eat it. So, when they come out, they say those are the dead now coming out, so someone says no." (FGD Nyawita, urban)*

*"A rich person will not park his car where the kumbe kumbe [flying termite] is coming out so that they can eat it; they will just pass. A poor person like me…I will just eat it." (FGD Ekwanda, rural)*

### 3.4. Perceptions of edible insect-based products

There was little prior knowledge and awareness about insect-based products in both the study regions. Most respondents indicated that they were learning about edible insect products for the first time from this study. It was apparent that local communities consumed edible insects in their natural rather than processed form. However, a few respondents were aware of edible insect-based processed products such as biscuits. Among respondents who agreed to taste the biscuits and cookies made with cricket-based ingredients, curiosity was the main driving factor as the majority indicated that it was their first time tasting an edible insect-based product. Furthermore, most of those who tasted the product were influenced by the attractive packaging and presentation in the familiar form of biscuits. Interestingly, respondents indicated that they believed the product was not harmful because of the assurance of quality checks on the package. Others indicated having no problem with consuming insect-based products since edible insects were considered normal food in the area. Other motivations for tasting the product included its attractive color, smell, and appearance. However, respondents who declined to taste the product cited inadequate information about the product as the main reason. Respondents also highlighted some barriers to the future consumption of edible insect-based products in the area. These included inadequate information about insect-based products, use of unfamiliar edible insects, hygiene and quality assurance during processing, and unrefined processing that left visible insect parts in the products. However, opinions on product labeling varied among respondents as some proposed the displaying of insect images on the packaging while others suggested the omission of insect images from packaging to avoid discouraging unsure potential consumers.

*"I wanted to know what ingredient was used to process it. It may be mealworms. I don't want to eat something I don't know. I have never eaten Onjiri. I asked what the ingredient was…" (FGD Chulaimbo, urban)*

*"I didn't know how it would taste...I didn't know that something like an insect can make biscuits, so I was just tasting to know how it tastes like." (FGD Daraja Mbili, urban)*

### 3.5. Suggestions to enhance consumption of edible insects

Respondents suggested potential strategies to overcome barriers to the consumption of edible insects as listed in Table 4. These include creating awareness among communities about edible insects, especially those not commonly consumed locally. They suggested providing detailed information including the benefits of edible insects, their safety as food, and debunking popular myths, and negative perceptions that hindered their consumption. Ensuring the continuous availability of edible insects was also considered crucial to increase their consumption. This was especially important for traditionally popular edible insects that have since become scarce, those not readily available, and those only available seasonally. Furthermore, collection of insects for consumption was reported to be extremely time-consuming and labor-intensive, discouraging their use as food. Hence processing edible insects into insect-based products, mass rearing, and elaborate marketing strategies were considered important for enhanced consumption. Furthermore, respondents perceived that research could mitigate risks linked to edible insects perceived to be destructive or harmful, such as queen termites and mealworms.

## Discussion

In this study, respondents demonstrated quite a substantial knowledge of edible insects, with flying termites, grasshoppers, soldier termites, and locusts being the most popular edible insects. Although knowledge of edible insects did not vary by market location, consumption was higher in rural markets. This could be because people in rural areas are more likely to consume edible insects since they are more available and accessible in the wild compared to urban markets due to the existence of less disturbed ecological systems in rural areas [28]. Place of residence may influence consumers' acceptance of edible insects, which could be attributed to existence of edible insect-based diets among rural communities [29,30]. Consumption of edible insects is traditionally passed down from one generation to another; hence acceptability may be higher in rural areas where traditional knowledge is still valued [31]. There were some differences in consumption experience and acceptability across different demographic groups, with non-natives and younger people reporting lower acceptability. These differences in acceptability between genders and age groups were mainly related to the ease or difficulty of harvesting and perceived benefits of edible insects. Contrary to our findings, studies in other settings have reported negative influences of age and gender on the acceptability of edible insects, with older and female consumers reported to be less likely to consume edible insects [32,33]. However, among individual respondents, nutritional value, taste, and familiarity with edible insects as a source of food were the most important determinants of edibility. Previous studies have reported edibility and choice of insects as food, to be determined by prior knowledge and familiarity with entomophagy, sensory appeal, availability, affordability, and convenience to consumers [34]. Furthermore, high nutritional value,

**Table 4. Suggestions to enhance consumption of edible insects and insect products.**

| Suggested strategies | Cricket | Grasshopper | Flying termite | Queen termite | Mealworm |
|---|---|---|---|---|---|
| Education, awareness, and sensitization | √ | √ | √ | √ | √ |
| Promotion and marketing strategies | √ | √ | √ | √ | √ |
| Enhancing availability | √ | √ | √ | √ | |
| Processing and packaging | √ | √ | √ | √ | |
| Insect rearing | √ | √ | √ | √ | |
| Research | √ | | √ | | √ |
| Risk mitigation | | | | √ | √ |

medicinal, and positive health benefits informed respondents' decisions to include edible insects in their diets. This agrees with earlier studies that highlighted perceived nutritional and health benefits of edible insects to increase consumers' acceptability [35]. We also found that cost and curiosity, which have been previously described as key predictors of willingness to try novel foods, influenced respondents' consumption of edible insects [36,37].

The main barriers to consumption of edible insects were related to unfamiliarity and lack of prior consumption experience which contributed to negative perceptions such as disgust among non-consumers. Consumer perception is a primary barrier to consumption of edible insects [38]. Cultural and religious beliefs, allergic reactions, and household socio-economic status presented further barriers to consumption, as has been reported by other studies [39,40]. Processing can offset key barriers to consumption of edible insects found in the study area, such as visual appearance, disgust, perishability, and seasonality [41,42]. Processed edible insects such as crickets and flying termites could be blended with other ingredients like wheat flour to make popular food products such as breads and buns [38]. This would enrich the nutritional value of major food items and address malnutrition, which is prevalent in poor households in developing countries. Previous studies have reported the use of edible insects in food items to improve nutrition, for instance among malnourished children in Kenya and Ghana [43,44].

We found little prior knowledge and awareness about insect-based products among respondents. Willingness to taste insect-based products was mostly driven by curiosity, with acceptability being influenced by attractive packaging, familiar product presentation in the form of biscuits and cookies, assurance of quality checks, and adequacy of information provided. Curiosity is an important factor influencing attempting novel insect-based products [45]. A similar study by Gorman et al. [46] showed that the presentation of different information impacted consumer perception of cricket-containing cookies. This strategy could be used to increase consumer acceptance of edible insect-based products. Other studies have encouraged the use of positive sensory exposures and tasting trials to increase familiarity with, and acceptance of edible insect consumption [47,48]. Developing diverse forms of insect-based products using different flavors and combinations of ingredients could also appeal to wider social groups within the community. For instance, the biscuits and cookies offered to respondents in this study could have appealed more to children than adults. Previous studies in different regions including Kenya and Denmark have reported good acceptability of cookies and other food items such as biscuits, porridge, and cereals containing insect-based ingredients, particularly among young children, who may be more receptive to trying new food products [43,49]. Finer crushing to reduce visible insect parts, which was a deterrent to some respondents, could also encourage the consumption of insect-based products. Use of edible insects in unrecognizable forms, with no visible parts, as ingredients in familiar food products has been shown to be preferred by consumers and to increase acceptance of insect-based products [45]. Elaborate marketing strategies that incorporate respondents' packaging and branding preferences, affordable costs, and ease of availability within local retail outlets could enhance the acceptability and consumption of insect-based products. Attractiveness of insect-based products may promote uptake, while labelling products as containing insects or including realistic packaging images may affect consumer's willingness to try the products [50].

Addressing knowledge, availability, and access barriers within communities could facilitate the consumption of edible insects. Indigenous knowledge and market availability of edible insects play an important role in their promotion and acceptance as an alternative source of food [51,52]. Adequate knowledge transfer could motivate consumption through shifting perceptions of disgust and of edible insects as pests or harmful creatures, thus supporting well-informed consumer decision-making [53]. Previous studies have suggested strategies to

encourage consumers' acceptance of edible insect-based products, including providing information on health benefits, involvement of social influencers or celebrities, and providing tasting experiences [54,55]. Although we found that some communities had existing traditional methods of harvesting edible insects outside their usual seasons of occurrence, these were inadequate to meet local consumption demands. Measures to ensure the sustained availability of edible insects through enhanced production, harvesting, and preservation techniques, while incorporating local knowledge, could ensure year-round supplies within communities [42,56]. Finally, to address hygiene, health, and quality concerns related to consumption of edible insects, such as microbial contamination in crushed insects, it is important for appropriate national food safety regulations to be developed and adhered to across the entire food value chain [57,58].

This study had some strengths and limitations. First, use of FGDs enabled rapid data collection from diverse respondents using few resources. Providing insect-based products for tasting also provided real-time feedback on acceptability based on actual sensory experiences. Our findings represent the views of a small sample of respondents from only two counties, hence are not generalizable to the larger population. However, these findings contribute to an understanding of attitudes and perceptions on edible insects' consumption that could inform future studies involving larger sample sizes and demographic groups to capture diverse attitudes within the broader population.

## Conclusions

This study demonstrates the need for interventions that promote public awareness and enhance the availability and accessibility of edible insects and insect-based products. Targeted community awareness and education could improve knowledge and facilitate a wider acceptability of edible insects, especially among young people and urban populations. Improving the availability of edible insects through supporting sustainable local production and designing appropriate harvesting and marketing techniques could improve local livelihoods and nutrition security. Our findings suggest that investing in research on important components of edible insects especially the nutritional and health benefits, rearing and harvesting techniques, methods of preparation, and potential health risks, could enhance local acceptability. Furthermore, involving local communities in knowledge co-creation and dissemination would be important for the sustainability of research-based interventions. Integrating local traditional knowledge and research into relevant food security policies and decision-making processes will also be crucial for promoting the consumption of edible insects. Future studies could focus on interventions that address the knowledge and attitudinal barriers to the consumption of edible insects, including cultural and religious beliefs. Furthermore, studies that consider factors related to community and individual socio-economic characteristics such as gender, age, location, and household income levels will be critical for designing appropriate edible insect promotion strategies.

## Supporting information

**S1 File. Data on Edible Insect_Liked_disliked.**
(XLS)

## Acknowledgments

The authors would like to thank all the respondents who participated in the focus group discussions (FGDs) for their time and for sharing their experiences.

## Author contributions

**Conceptualization:** Emmah Owidi, Fanuel Kawaka.

**Data curation:** Gilbert Asoka, Eric Waga.

**Formal analysis:** Emmah Owidi, Gilbert Asoka, Eric Waga.

**Funding acquisition:** Fanuel Kawaka.

**Investigation:** Emmah Owidi, Eric Waga, Alfred Ochieng'.

**Methodology:** Emmah Owidi, Gilbert Asoka, Eric Waga, Alfred Ochieng', Fanuel Kawaka.

**Project administration:** Fanuel Kawaka.

**Supervision:** Fanuel Kawaka.

**Writing – original draft:** Emmah Owidi.

**Writing – review & editing:** Gilbert Asoka, Alfred Ochieng', Fanuel Kawaka.

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
