## [Decision Letter · Decision Letter 0]

18 Dec 2024

PONE-D-24-55547Consumer attitudes and perceptions on consumption of edible insects among communities in western KenyaPLOS ONE

Dear Dr. Kawaka,

Thank you for submitting your manuscript to PLOS ONE. After careful consideration, we feel that it has merit but does not fully meet PLOS ONE’s publication criteria as it currently stands. Therefore, we invite you to submit a revised version of the manuscript that addresses the points raised during the review process.

We look forward to receiving your revised manuscript.

Kind regards,

António Raposo

Academic Editor

PLOS ONE

2. During the internal evaluation of the study we have noted that potentially identifying information was presented within the manuscript text. Please clarify whether the names used in your manuscript are pseudonyms. If not, please remove or replace by pseudonyms.

3. We note that your Data Availability Statement is currently as follows: [All relevant data are within the manuscript and its Supporting Information files.] Please confirm at this time whether or not your submission contains all raw data required to replicate the results of your study. Authors must share the “minimal data set” for their submission. PLOS defines the minimal data set to consist of the data required to replicate all study findings reported in the article, as well as related metadata and methods (https://journals.plos.org/plosone/s/data-availability#loc-minimal-data-set-definition).

Additional Editor Comments (if provided):

Reviewers' comments:

Reviewer's Responses to Questions

**Comments to the Author**

1. Is the manuscript technically sound, and do the data support the conclusions?

Reviewer #1: Yes

Reviewer #2: Yes

2. Has the statistical analysis been performed appropriately and rigorously? 

Reviewer #1: Yes

Reviewer #2: N/A

3. Have the authors made all data underlying the findings in their manuscript fully available?

Reviewer #1: Yes

Reviewer #2: Yes

4. Is the manuscript presented in an intelligible fashion and written in standard English?

Reviewer #1: Yes

Reviewer #2: Yes

5. Review Comments to the Author

Reviewer #1: The manuscript addresses a valid research question and aligns with the scope of the journal. Additionally, it explores an important topic by investigating local communities' attitudes and perceptions towards various edible insects and insect-based products.

To enhance the development and structure of the text, I would like to offer a few comments, questions, and suggestions:

- The title and abstract clearly and concisely indicate the focus of the manuscript.

2.4 Data Analysis

- It is essential to ensure that all text is consistent in color, as this can be identified in the software reference. I recommend addressing this issue.

3. Results

- I suggest that examples of participants' statements be presented in the same format as in Example 3.2, with proper indentation, spacing, and enclosed in quotation marks.

- How were the insects consumed? Were their forms of preparation or presentation described (e.g., dried, fresh, crushed)? Providing this detail would add clarity and context to the study.

Discussion

- I recommend organizing the discussion in a structured manner, following the approach used in the introduction and methods.

- Consider discussing the risks to food and nutritional security associated with encouraging this production and consumption. Emphasize the potential health risks involved and propose measures to enable users to produce and consume these products safely, avoiding contamination during collection, market availability, and general consumption.

- I suggest including additional references to support the data presented in the second paragraph of this section.

- Furthermore, I recommend incorporating future research perspectives along with the study's limitations in a separate section following the discussion.

Conclusion

- Alongside summarizing the main findings, it is important to outline prospects for future research and provide an overall conclusion of the study, while also detailing the results individually.

References

- I recommend including more recent references, as it was observed that 60% of the references are over five years old.

Reviewer #2: This article offers interesting insights into the acceptability and consumption of edible insects in western Kenya.

The paper is limited to FGDs in only two countries, Kisumu and Vihiga. While informative, this narrow geographical focus may not capture the diversity of attitudes across Kenya or even other regions within western Kenya. A more comprehensive study with broader regional representation would enhance the generalizability of the findings.

While the study involved 59 respondents, the demographic representation might not fully capture the diverse attitudes within the broader population. Increasing the sample size and ensuring a wider range of ages, education levels, and socio-economic backgrounds could enhance the robustness of the conclusions drawn.

The manuscript relies on conventional content analysis without detailing how themes were derived, which may affect transparency and reproducibility. A more rigorous analytical framework, such as grounded theory or thematic analysis with inter-coder reliability checks, could strengthen the study’s validity.

There is no mention of efforts to address potential biases in the FGDs, such as groupthink or interviewer influence, which could have affected the data.

Although the research mentions the gender distribution of participants, it does not delve deeply into how attitudes toward edible insects may vary by gender or other demographic factors such as age, education, or income levels. This omission weakens the ability to design tailored interventions.

The paper briefly mentions seasonality as a factor limiting insect consumption but does not explore strategies to mitigate this challenge. For instance, addressing preservation methods or year-round farming could provide actionable insights.

The paper notes differences in acceptance between rural and urban areas but doesn't explore the underlying reasons for these differences in depth. Additional qualitative data could provide insights into cultural, economic, or educational factors that influence attitudes in these contrasting environments.

Although the study suggests that improving the availability of edible insects could enhance local livelihoods, it does not evaluate the economic feasibility or market dynamics of scaling up insect production. Addressing these aspects would add practical value to the recommendations.

The suggestion to integrate traditional knowledge into food security policies is crucial but underdeveloped. More concrete examples of how traditional practices can inform modern interventions would make this recommendation more actionable.

The study repeats the need for promoting public awareness and improving availability of edible insects in multiple sections. Consolidating these ideas into a coherent discussion would improve clarity and avoid redundancy.

The study does not compare its findings with other regions or countries where edible insect consumption is common. Including such comparisons could enrich the discussion and highlight best practices or transferable strategies.

While the study mentions curiosity and familiarity as factors influencing willingness to try insect-based products, it does not explore innovative ways to incorporate edible insects into widely accepted foods (e.g., snacks, flour). Addressing this could appeal to urban populations and younger demographics

6. PLOS authors have the option to publish the peer review history of their article (what does this mean? ). If published, this will include your full peer review and any attached files.

**Do you want your identity to be public for this peer review?** For information about this choice, including consent withdrawal, please see our Privacy Policy .

Reviewer #1: **Yes: ** Marcela Gomes Reis

Reviewer #2: **Yes: ** M. João Reis Lima

---

## [Author Response · Author response to Decision Letter 1]

14 Jan 2025

Editor The authors should submit the following data behind the means, standard deviations and other measures reported;

The authors have submitted all the raw data required to replicate the results of your study

Reviewer #1:

Section Reviewer comments Author response Document page and line number

Title The title and abstract clearly and concisely indicate the focus of the manuscript Thank you for your comment.

Data analysis It is essential to ensure that all text is consistent in color, as this can be identified in the software reference. Thank you for the correction. We have harmonized the text color. Page 3; line 119

Results I suggest that examples of participants' statements be presented in the same format as in Example 3.2, with proper indentation, spacing, and enclosed in quotation marks. Thank you for the suggestion. We have harmonized the format as suggested. Page 6; Lines 189-193

How were the insects consumed? Were their forms of preparation or presentation described (e.g., dried, fresh, crushed)? Providing this detail would add clarity and context to the study. Thank you for this important comment. We have included information on forms of preparation. Page 4; Line 141-142

Page 8; Lines 256-258

Discussions I recommend organizing the discussion in a structured manner, following the approach used in the introduction and methods. Thank you for this important comment. We have reorganized the discussion section. Pages 10-11; lines 312, 318, 329

Consider discussing the risks to food and nutritional security associated with encouraging this production and consumption. Emphasize the potential health risks involved and propose measures to enable users to produce and consume these products safely, avoiding contamination during collection, market availability, and general consumption. Thank you for this important comment. We have included information on health risks involved. Page 11; lines 353-356

I suggest including additional references to support the data presented in the second paragraph of this section. Thank you for this important comment. We have included additional references to support our discussion. Page 11; Lines 326-343

Furthermore, I recommend incorporating future research perspectives along with the study's limitations in a separate section following the discussion. Thank you for this important comment. We have included information on study limitations and future research perspectives. Page 12-13; lines 400-407

Conclusion Alongside summarizing the main findings, it is important to outline prospects for future research and provide an overall conclusion of the study, while also detailing the results individually. Thank you for this important comment. We have revised the conclusions to include future research prospects and an overall conclusion. Page 13; Lines 398-416

References I recommend including more recent references, as it was observed that 60% of the references are over five years old Thank you for this important comment. We have included recent references in the introduction and discussion sections. Page 1-2; Lines 45-67

Page 11-12; Lines 309-386’ All the new references have been listed and highlighted in yellow

Reviewer #2:

Reviewer comments Author response Document page and line number

The paper is limited to FGDs in only two countries, Kisumu and Vihiga. While informative, this narrow geographical focus may not capture the diversity of attitudes across Kenya or even other regions within western Kenya. A more comprehensive study with broader regional representation would enhance the generalizability of the findings. Thank you for this important comment. This was a pilot study that focused on providing preliminary findings to inform a larger future study to capture diverse attitudes within the broader population. This is recognized as a study limitation. Page 11; line 354-361

While the study involved 59 respondents, the demographic representation might not fully capture the diverse attitudes within the broader population. Increasing the sample size and ensuring a wider range of ages, education levels, and socio-economic backgrounds could enhance the robustness of the conclusions drawn. Thank you for this important comment. The focus of this study was to provide preliminary findings to inform a larger future study to capture diverse attitudes within the broader population. This is recognized as a study limitation. Page 11; line 354-361

The manuscript relies on conventional content analysis without detailing how themes were derived, which may affect transparency and reproducibility. A more rigorous analytical framework, such as grounded theory or thematic analysis with inter-coder reliability checks, could strengthen the study’s validity. Thank you for this important comment. We have revised the section and included more details on the procedure used for analysis, which was actually thematic in nature. Page 3; Lines 120-132

There is no mention of efforts to address potential biases in the FGDs, such as groupthink or interviewer influence, which could have affected the data. Thank you for this important comment. We have included measures we used to control against potential bias during the FGDs. Page 3; Lines 102-111

Although the research mentions the gender distribution of participants, it does not delve deeply into how attitudes toward edible insects may vary by gender or other demographic factors such as age, education, or income levels. This omission weakens the ability to design tailored interventions. Thank you for this important comment. We found some differences in knowledge and consumption among women/men; younger/older and lower/higher socio-economic levels. We have revised the results to include those different perspectives. Page 8; Lines 230-233

Page 7; Lines 219-223

Page 8: Lines 240-255

Page 9; Lines 274-280

Page 12; Lines 361-368

The paper briefly mentions seasonality as a factor limiting insect consumption but does not explore strategies to mitigate this challenge. For instance, addressing preservation methods or year-round farming could provide actionable insights. Thank you for this important comment. We have included measures to address seasonality and supply challenges. Page 12; Lines 389-391; 393-396

The paper notes differences in acceptance between rural and urban areas but doesn't explore the underlying reasons for these differences in depth. Additional qualitative data could provide insights into cultural, economic, or educational factors that influence attitudes in these contrasting environments. Thank you for this important comment. We agree with this suggestion. We did not find clear underlying reasons for differences in acceptability between rural and urban areas, except for the existence of edible insects in traditional diets. We have included some discussion on potential reasons for higher consumption in rural areas. Page 10-11; Lines 312-321

The suggestion to integrate traditional knowledge into food security policies is crucial but underdeveloped. More concrete examples of how traditional practices can inform modern interventions would make this recommendation more actionable. This has been included in the manuscript as recommendations to be considered for national and county food policies- e.g. Kenya national nutrition action plan

Page 14; Lines 623-625

The study repeats the need for promoting public awareness and improving availability of edible insects in multiple sections. Consolidating these ideas into a coherent discussion would improve clarity and avoid redundancy. Thank you for this comment. We have harmonized the discussion on public awareness and improving availability to reduce redundancy as suggested. Lines 476-477; 612-616

The study does not compare its findings with other regions or countries where edible insect consumption is common. Including such comparisons could enrich the discussion and highlight best practices or transferable strategies. The revised manuscript has included comparison with other studies in the discussion Lines 432-449; 481-534; 537-545; 558-568

While the study mentions curiosity and familiarity as factors influencing willingness to try insect-based products, it does not explore innovative ways to incorporate edible insects into widely accepted foods (e.g., snacks, flour). Addressing this could appeal to urban populations and younger demographics Thank you for this important comment. We have included a discussion on curiosity and familiarity and incorporating edible insects in popular food items. Page 11-12; Lines 332-370

---

## [Decision Letter · Decision Letter 1]

21 Jan 2025

Consumer attitudes and perceptions on consumption of edible insects among communities in western Kenya

PONE-D-24-55547R1

Dear Dr. Kawaka,

We’re pleased to inform you that your manuscript has been judged scientifically suitable for publication and will be formally accepted for publication once it meets all outstanding technical requirements.

Kind regards,

António Raposo

Academic Editor

PLOS ONE

Additional Editor Comments (optional):

Reviewers' comments:

Reviewer's Responses to Questions

**Comments to the Author**

1. If the authors have adequately addressed your comments raised in a previous round of review and you feel that this manuscript is now acceptable for publication, you may indicate that here to bypass the “Comments to the Author” section, enter your conflict of interest statement in the “Confidential to Editor” section, and submit your "Accept" recommendation.

Reviewer #1: All comments have been addressed

Reviewer #2: All comments have been addressed

2. Is the manuscript technically sound, and do the data support the conclusions?

Reviewer #1: Yes

Reviewer #2: Yes

3. Has the statistical analysis been performed appropriately and rigorously? 

Reviewer #1: Yes

Reviewer #2: Yes

4. Have the authors made all data underlying the findings in their manuscript fully available?

Reviewer #1: Yes

Reviewer #2: Yes

5. Is the manuscript presented in an intelligible fashion and written in standard English?

Reviewer #1: Yes

Reviewer #2: Yes

6. Review Comments to the Author

Reviewer #1: The authors have made changes to all the issues raised, I thank them for their comments and suggest that the manuscript be accepted.

Reviewer #2: Considerind the auhor´s answers, I agree that the manuscript can be published after reviewing the modifications

7. PLOS authors have the option to publish the peer review history of their article (what does this mean? ). If published, this will include your full peer review and any attached files.

**Do you want your identity to be public for this peer review?** For information about this choice, including consent withdrawal, please see our Privacy Policy .

Reviewer #1: **Yes: ** Marcela Gomes Reis

Reviewer #2: **Yes: ** M. João Reis Lima

---

## [Editor Report · Acceptance letter]

PONE-D-24-55547R1

PLOS ONE

Dear Dr. Kawaka,

I'm pleased to inform you that your manuscript has been deemed suitable for publication in PLOS ONE. Congratulations! Your manuscript is now being handed over to our production team.

Kind regards,

on behalf of

Dr. António Raposo

Academic Editor

PLOS ONE